# Functional Variability in Team-Handball Players during Balance Is Revealed by Non-Linear Measures and Is Related to Age and Expertise Level

**DOI:** 10.3390/e22080822

**Published:** 2020-07-27

**Authors:** Carla Caballero, David Barbado, Tomás Urbán, Juan Antonio García-Herrero, Francisco J. Moreno

**Affiliations:** 1Department of Sport Sciences, Sport Research Center, Miguel Hernandez University of Elche, 03202 Elche, Spain; ccaballero@umh.es (C.C.); dbarbado@umh.es (D.B.); t.urban@umh.es (T.U.); 2Department of Didactics of Musical, Plastic and Corporal Expression, Faculty of Education, University of Salamanca, 37008 Salamanca, Spain; gherrero@usal.es

**Keywords:** postural control, team-handball throwing, movement variability, experience, age

## Abstract

Postural control is considered a key variable in team sports, such as handball, which require abilities strongly related to balance. However, postural control and its relationship to the performance of handball skills according to the players’ skill level and age has not been evaluated to date. This study analyzes the relationship between balance ability and team-handball performance according to age and expertise, applying a non-linear approach to balance assessment. Postural control from 114 male team-handball players was analyzed through the center of pressure (COP) during a balance task. Sport performance was measured by the accuracy and speed in throwing. Expert players threw faster, but not more accurately than recreational players. Balance performance was better for 18+ players (older than 18 years old) than those U12 (under 12 years old), but no differences were found according to their skill level. Players who threw with less accuracy showed slower COP velocity during the balance task and their moves were less irregular. Players who threw faster displayed more irregular and less auto-correlated COP movements. In conclusion, experienced team-handball players exhibited better balance performance, and this seems to be related to the maturation of the motor system more than to sport performance level. Nevertheless, non-linear measures of COP excursion revealed an exploratory behavior during balance in expert players, exhibiting more motion adjustments to reduce motor output error. Traditional variables measuring balance performance did not show sensitivity to this motor control process. A non-linear approach to balance assessment revealed functional variability during balance as an intrinsic characteristic of individuals’ motor control according to age and skill level.

## 1. Introduction

Studies have indicated that sport training improves sensorimotor performance and postural control [1,2]. Thus, the level of sport training should be related to the level of postural control. In general, apart from speed and explosive strength, an improvement of balance should be considered as one of the key features of agility improvement [3]. However, the balance influence on several athletic skills in different sports has not always been observed [4].

Studies that have analyzed sports like gymnastics, soccer, or golf have suggested that practitioners of these sports display a higher balance control than their less proficient counterparts or non-athletes [5,6,7], but no differences have been found in sports like judo or surfing [8,9]. In addition, studies that have tried to relate the balance ability and performance level of specific sport tasks showed no clear differences. Behm, Wahl [10] indicated that, in hockey skating, stability may be associated with skating speed in young players, but this relation is not seen in players over 19 years old. Mononen, Konttinen [11] studied the relationship between shooting accuracy and postural balance, and they indicated that this relation was only present at the inter-individual level. In golf, a study showed a significant relationship between dominant and non-dominant leg balance and performance after a chip shot [7]. However, in other sports like baseball, no significant correlation was found between balance performance (unilateral stance eyes open or unilateral stance eyes closed) and performance (pitch velocity) [12].

Some studies have reported significant differences between groups with different levels of expertise only on unstable surfaces [8,13]. It seems that the balance task should be sensitive enough to detect subtle differences between groups of healthy people, and static balance tasks may not meet this requirement [14]. Zemkova [15] found no significant differences in postural stability between athletes of different specializations and physically active individuals during standing in a standard upright position. Another possible reason for the previous conflicting results may be the variable used to assess balance performance. Most studies have used average scalar parameters, describing the sway and the dispersion or area during a given time to assess postural control [16,17]. However, there is no agreement about the optimal selections of discriminative sway parameters [18]. Recently, non-linear measures have been used and they may be helpful to quantify how motor behavior changes over time [19,20]. Some authors indicate that postural control efficiency is reflected in the dynamic characteristics of sway patterns, and that differences in those characteristics reflect differences in postural control that are better identified by non-linear parameters [19].

Team-handball is a sport that requires abilities (e.g., throwing, jumping, hopping and skipping) that are strongly related to balance [21]. In addition, balance performance has been related to sport injury risk, and it has been considered as one of the limiting factors of performance in numerous sports [15]. Zemkova [15] suggested that, in this type of sport, loss of balance during fast side-to-side movements may contribute to knee injuries. However, even though physical fitness attributes of team-handball players have been analyzed in previous studies [22,23], to the best of our knowledge, postural control and its relationship with team-handball skill performance has not been assessed to date.

Owing to the unclear results reported by the literature, the aim of this study was to assess the relationship between balance ability and sport performance (defined as throwing speed and accuracy) in team-handball players of different ages and levels of expertise, with the aim of analyzing the relevance of balance control in throwing performance.

## 2. Method

### 2.1. Participants

One hundred and fourteen right-handed male team-handball players with different levels of experience and ages took part in this study (see Table 1 for demographic information). Sixty of them were expert players recruited from the Spanish Handball Federation, who played in national or international tournaments and trained at least four times per week. Fifty-four players were categorized as recreational players as they satisfied the following requirements: (i) to train no more than two times per week and (ii) to not take part in any national or international competition. None of the 114 participants had previous experience in the balance task used in this study.

Written informed consent was obtained from each participant prior to the experiment. For both U12 (under 12 years old) and U16 (under 16 years old) groups, written informed consent was signed by the players’ guardians. Data were treated anonymously. The experimental procedures used in this study were in accordance with the Declaration of Helsinki and were approved by a University Office for Research Ethics (Ref: DPS.FMH.01.16).

### 2.2. Experimental Procedure and Data Collection

Participants performed two different protocols:

1) Balance task procedure.

Participants were asked to stand “as still as possible” [24] on an unstable surface for one minute. The unstable surface consisted of a rigid wooden platform (diameter: 55 cm) affixed to the flat surface of a polyester resin hemisphere (diameter of hemisphere: 35 cm; height of the platform relative to the bottom of the hemisphere: 12 cm). Their feet were placed shoulder-width apart with their hands resting on their hips. The foot position was such that the line between their heels coincided with the medio-lateral axis of the platform.

In order to assess postural stability, center of pressure (COP) was estimated by a force platform (Kistler 9286AA0) at 20 Hz [24].

2) Team-handball throwing procedure.

Before testing, all participants completed a 10 min warm-up consisting of jogging, stretching, and throwing at a submaximal velocity. During the protocol, each player performed 30 throws distributed into three sets of 10 throws. Participants rested for 30 s between sets and 5 s between throws. They were instructed to “throw at the highest speed and with the highest accuracy possible, aiming at the target” [25]. The balls (International Handball Federation size 1, 2, or 3 depending on the players’ age; Molten, Japan) were given to the participant one by one. The target was placed in the upper right corner of the goal (Figure 1).

To assess the maximum ball speed during each throw, a Sports Radar SR3600 (Homosassa, FL, USA) was placed behind the players pointing at the target (Figure 1). The goal and surrounding areas were covered with a net. The goal was filmed during the evaluation with a digital camera (50 Hz sampling frequency; HDR-SR8E; Sony, Tokyo, Japan) to establish the point of impact of the ball when it touched the net for each throw. The camera was placed at a height of 3 m above the floor, pointing at the target (Figure 1).

## 3. Data Analysis

1) Balance task procedure.

Before any variable computation, COP signal was processed according to the protocol by Caballero, Barbado [24]. Postural performance was assessed using traditional bivariate COP-based measures combining the anterior–posterior (AP) and medial–lateral (ML) displacement trajectories: bivariate variable error (BVE) and mean velocity magnitude (MVM). These variables were calculated over the signal using the approach by Prieto, Myklebust [26]; that is, BVE was measured as the average value of the absolute distance to each participant’s own midpoint, and MVM was measured as the average velocity of the COP [27].

The variables used to assess the variability structure of displacement and velocity of the COP time series were fuzzy entropy (FE_D_ − FE computed over the displacement of the COP time series, and FE_V_ − FE computed over the velocity of the COP time series) and detrended fluctuation analysis (DFA_D_ and DFA_V_). The variables were calculated over the resultant distance (RD) of the COP time series displacement and velocity data, calculated according to Caballero, Barbado [28].

FE typically returns values that indicate the degree of irregularity in the signal. It was calculated according to Chen, Wang [29]. This measure computes the repeatability of vectors of length m and m + 1 that repeat within a tolerance range of r of the standard deviation of the time series. Thus, low FE values represent a high repeatability within the time series as similar vectors of length m remain similar for the next + 1 points. Conversely, high FE values represent a low repeatability within the time series as similar vectors of length m do not remain similar for the next + 1 points. To calculate this measure, the following parameter values were used: vector length, m = 2; tolerance window, r = 0.2*SD; and gradient, n = 2 [29]. Although there are many entropy measures, FE was used because it displays a stronger relative consistency, less dependency on data length, free parameter selection, and more robustness to noise than other measures like sample entropy [30].

DFA represents a modification of classic root mean square analysis with random walk to evaluate the presence of long-term correlations within a time series using a parameter referred to as the scaling index, α [31]. The scaling index α corresponds to a statistical dependence between fluctuations at one time scale and those over multiple time scales [32]. This variable was computed according to the procedures by Peng, Havlin [31], using the window range 4 ≤ n ≤ N/10 to maximize the long-range correlations and reduce errors incurred in estimating α [33]. Different values of α indicate the following: α > 0.5 implies persistence in position (the trajectory tends to remain in its current direction) and α < 0.5 implies anti-persistence in position (the trajectory tends to return to where it came from) [34].

2) Team-handball throwing procedure.

The ball speed (km/h) was computed as the mean maximum speed of the ball. To compute the accuracy of the throw (m), the mean radial error (MRE) of the position of the ball when it touched the net was calculated as the average absolute distance to the center of the target (see Figure 1). Video digitalization of the ball impact was computed with a Matlab (version 7.11; Mathworks, Natick, MA, USA) routine for the calculation of real-space Cartesian coordinates for computing the distance from the ball to the center of the target.

## 4. Statistical Analysis

Normality was evaluated using the Kolmogorov–Smirnov test with the Lilliefors correction. Two-way independent measures analyses of variance (ANOVA) were carried out to assess between group differences in all parameters, with *age* (U12, U16, and +18) and *skill level* (expert and recreational) being the between-subjects factors. Statistical significance was set at *p* < 0.05. Bonferroni adjustment was performed to ascertain differences between balance and sport performance for age and expertise factors. Partial eta squared (${µ}_{p}^{2}$) was calculated as a measure of effect size to provide a proportion of the overall variance attributable to the factor. Values of effect size ≥0.64 were considered strong, around 0.25 moderate, and ≤0.04 small [35]. Pearson product moment correlation coefficients were calculated to assess the relationships between variables.

## 5. Results

Regarding team-handball performance variables, significant differences were found only in ball speed between age ranges and expertise level (Table 2). The results showed clearly that expert players threw faster than recreational players, finding differences between expertise levels for all age ranges (*p* < 0.001). 

Balance performance was significantly better (lower BVE values) for players over 18 years old (18+) than for those under 12 years old (U12) (Table 2), but these differences were found only in the expert players group. No differences were found for MVM values according to the age ranges. Focusing on the structure of COP variability, FE_V_ was the only variable that displayed significantly different values. Recreational players U12 displayed lower FE_V_ values than players under 16 years old (U16) and 18+. This variable also showed significant differences according to the skill levels, as well as DFA_D_ (Table 2). Expert players showed higher FE_V_ and lower DFA_D_ values than recreational players, but these differences were only found in the U12 group.

Table 3 shows the correlation between all the variables for the whole cohort. Throwing accuracy and ball speed were not correlated. Regarding balance and team-handball performance, MRE was negatively correlated with MVM and FE_D_, while ball speed was positively related to FE_D_ and FE_V_ and negatively related to DFA_D_. Players who threw with less accuracy (higher MRE) moved slower during the balance task (lower MVM) and their moves were less irregular (lower FE_D_). On the other hand, players who threw faster (higher ball speed) displayed COP movements that were more irregular (higher FE_D_ and FE_V_) and less auto-correlated (lower DFA_D_).

When the sample was split by age range, the correlation between MRE and FE_D_ with MVM appeared only in the 18+ group. The relation between ball speed and FE_D_ disappeared, while the relationship between FE_V_ and DFA_D_ appeared only in the U12 group, showing the same trend described above (Table 4). Focusing on the skill level (Table 5), in the expert group, ball speed was negatively correlated with BVE. Players who threw faster (higher ball speed) displayed a lower error in their balance task performance (lower BVE). In this group, no more correlations between team-handball performance and balance were found. Regarding the recreational group, throwing performance was not related to balance performance. However, players who threw faster (higher ball speed) displayed a more irregular (higher FE_V_) and less auto-correlated (lower DFA_V_) COP velocity signal during the balance task.

## 6. Discussion

Previous studies have shown unclear results in the relationship between balance control and sport performance, and in team-handball, this relationship had not yet been assessed. In this study, the relationship between balance ability and sport performance of the team-handball throw in team-handball players of different ages and levels of expertise was addressed.

The main throwing performance variable that showed differences according to age range and skill level was the ball speed. The 18+ expert group threw significantly faster than any other group, with values for absolute ball speed (72 km/h) consistent with those reported in previous studies using a standing throwing action (70.2 km/h, Granados, Izquierdo [36]; and 77 km/h, van den Tillaar and Ettema [37]). In addition, the differences found in ball speed values between the age ranges and between the expert and recreational groups are similar to previous studies that used the same standing throwing action [38]. However, MRE values did not differ between age ranges or skill level groups. These findings are not coherent with the results observed in Rousanoglou, Noutsos [39], which showed that expert team-handball players displayed greater accuracy than novice players. In the present study, the recreational group included participants who trained one or two days per week, but did not compete, while in Rousanoglou, Noutsos’s [39] study, the group with a lower skill level included novice players who had completed a four-month team-handball course. This training background of the recreational groups in the current study may explain the smaller differences in throwing accuracy between the expert and recreational groups. Additionally, the instruction provided to “throw at the highest speed and with the highest accuracy possible” could have encouraged players to adapt the velocity in their throws to maintain acceptable accuracy levels [40].

Concerning balance performance, there were no significant results according to the expertise level, with these results being in accordance with those of Mohamed, Vaeyens [41]. However, there were differences according to age ranges regarding to BVE. Players in the 18+ group displayed lower BVE than those in the other groups. Yet, these differences were only found in the expert group. In other sports, no significant differences in balance performance were found according to expertise [8,9]. It seems that balance control is more related to age than skill level. This may be owing to the significant increases in vestibular function and to the better management of afferent sensory systems along maturation [42].

The variables that described the variability structure of the COP (DFA and FE) during the balance task were more sensitive to the differences not only between age ranges, but also between skill levels. The players who threw faster displayed higher irregularity and lower auto-correlation in their COP signal. According to some studies, a COP signal with these features indicates a higher number of movement adjustments in a balance task [20,28,43]. The authors support the idea that these features can be interpreted as an exploratory behavior that allows participants to perform more motion adjustments to reduce motor output error, achieving better performance [43]. Furthermore, something important to take into consideration is that these non-linear variables showed higher sensitivity to the skill level differences than balance performance variables. This indicates that this kind of variable may be useful to provide information about changes in postural control that linear variables cannot identify, because they quantify how motor behavior changes over time [19].

Regarding correlational analysis, the team-handball performance variables (accuracy and ball speed) were not related to each other. Some studies have addressed the relationship between throwing accuracy and speed in team-handball players according to instructions, and it seems that this relationship depends on the instructions and the skill level [37,40]. In this paper, the instruction given to the participants was to “throw at the highest speed and with the highest accuracy possible, aiming at the target”, being similar to the one used by van den Tillaar and Ettema [37]. No correlation was found by these authors either. Yet, no relationship between performance variables was found when only the recreational group was analyzed. This could be explained by the lack of a goalkeeper during the throws, because it seems that the existence of the goalkeeper had a significantly greater effect on throwing performance on a less experienced group compared with a more experienced group [44].

Focusing on the relationship between team-handball variables and balance performance, the accuracy of the throws of the whole sample showed a slight positive correlation with MVM. Players who had better accuracy in their throws moved faster during the balance task. MVM has been interpreted as an index of the amount of corrections needed to adjust the COP location, increasing neuromuscular effort resulting from participant’s exploratory behaviors [28]. These results seem to be consistent with previous studies in other sports like hockey skating, shooting, and golf [7,10,11]. However, ball speed correlated negatively with BVE in the expert group. Thus, it may be concluded that, in this study, there is no clear relationship between balance and team-handball performance.

Finally, the correlational results indicated that there was a relationship between team-handball performance variables and the variability structure of the COP. On the one hand, players who had better accuracy displayed a more irregular COP excursion. On the other hand, players who threw faster showed COP movements that were more irregular (higher FE_D_ and FE_V_) and less auto-correlated (lower DFA_D_). FE and DFA assess the extent to which further motor behavior is dependent on previous fluctuations. Less dependence on previous behavior (lower regularity and long-range auto-correlation, respectively) has been interpreted as higher flexibility to perform motion adjustments [45]. This means that players who performed their throws better showed a more exploratory behavior performing the balance task and, therefore, a greater adaptive capacity of the central nervous system (CNS) over longer time scales [43].

It seems that the strength of the correlations between team-handball performance and the variability structure of COP changed depending on the age ranges or on the skill levels (see Table 4 and Table 5). Nevertheless, entropy and auto-correlation values measured during the balance task showed a stronger relationship with sport performance than the traditional tools used to assess balance. According to these results, the matter of using this type of variable, which assesses how the motor output changes over time, reflecting how the CNS exploits all the possible motor outputs and corrections [28,43] to address the importance that balance has in sports, should be pointed out.

The current study presents some limitations. To reduce the effect of fatigue, participants performed a reduced number of throws. A larger number of trials measured in different sessions would have provided supplementary information about performance consistency. The absence of a balance task familiarization can be seen as another limitation of the study, but the lack of experience in the balance task was required to avoid specific learning effects in the balance task.

In conclusion, more experienced team-handball players exhibited better balance performance (i.e., lower sway), but this increased motor control related to age seems to be owing to the maturation of the motor system more than to their level in sport performance. The relationship between expertise and balance behavior has been found only in the structure of COP variability during the balance task. Expert players exhibited a more irregular and less auto-correlated variability, interpreted as an exploratory behavior, and more motion adjustments to reduce motor output error. These results indicate the significance of using non-linear variables because they provide information about functional variability and movement dynamics during balance tasks, revealing the intrinsic characteristics of the individual’s motor control. Individuals’ intrinsic features seem to be related not only to the maturation of the practitioners, but also to the sport performance. These results are related to previous studies that have related the complexity of the variability structure to a functional exploratory behavior and the ability to adapt.

### Practical Implications

This paper supports the idea that a non-linear approach is a more sensitive option than the traditional approach to quantify and reveal the athlete’s functional variability during balance tasks. This functional variability is an intrinsic characteristic of handball players’ motor control modulated by age and skill level, which is related to the ability to explore the environment in order to optimize motor performance. Practitioners are encouraged to design practice conditions that facilitate functional variability in players’ motor behavior so they will be able to explore all the possible motor solutions to find the best option and achieve a good sport performance. Non-linear measures of variability can be a practical tool to detect the presence of functional fluctuations related to better performance.

## Figures and Tables

**Figure 1 entropy-22-00822-f001:**
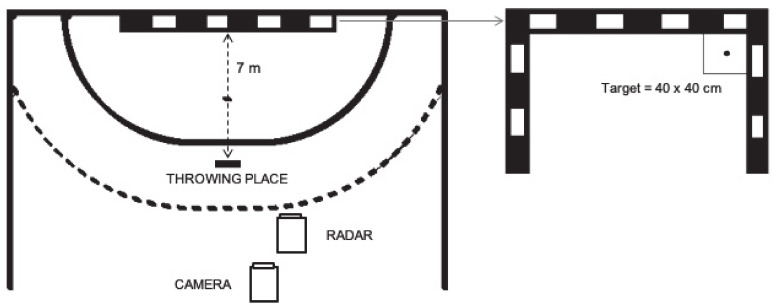
Handball field and instrument distribution.

**Table 1 entropy-22-00822-t001:** Demographic information from the participants who took part in the study.

	Handball Players			
	Expert		Recreational	
	N	Age (M ± SD)	N	Age (M ± SD)
**U12**	20	12.3 ± 0.6	22	12.4 ± 0.7
**U16**	16	15.9 ± 0.4	12	15.7 ± 0.3
**18+**	24	23.3 ± 3.4	20	26.5 ± 5.1

U12 = under 12 years old; U16 = under 16 years old; 18+ = older than 18 years old.

**Table 2 entropy-22-00822-t002:** Mean (SD) of team-handball throw and balance parameters according to the age and the expertise level of the team-handball players. Main effects and interaction effects (F, *p*-values, and effect size) are also presented.

		**U12**		**U16**		**18+**		*Age Effect*		
*Team-Handball Throw*										
**MRE**	**Expert**	0.349	(0.085)	0.391	(.061)	0.378	(0.107)^AB^	F_2,1013_ = 0.703	*p* = 0.497	ƞp2 = 0.013
	**Recreational**	0.405	(0.126)	0.368	(.077)	0.422	(0.089)^A^			
	*Expertise Effect*	F_1,113_ = 1.896			*p* = 0.171	ƞp2 = 0.017		*Interaction*		
								F_2,113_ = 1.494	*p* = 0.229	ƞp2 = 0.027
**Ball Speed**	**Expert**	59.48	(6.09)	64.70	(6.91)	71.58	(7.27)	F_2,113_ = 21.088	*p* < 0.001	ƞp2 = 0.281
	**Recreational**	44.71	(5.65)*	49.82	(6.69)*	51.23	(7.06)*			
	*Expertise level*	F_1,113_ = 170.775			*p* < 0.001	ƞp2 = 0.613		*Interaction*		
								F_2,113_ = 2.337	*p* = 0.101	ƞp2 = 0.041
*Balance Task*										
**BVE**	**Expert**	0.021	(0.005)	0.019	(0.004)	0.016	(0.004)^A^	F_2,113_ = 5.769	*p* = 0.004	ƞp2 = 0.097
	**Recreational**	0.020	(0.007)	0.017	(0.004)	0.018	(0.005)			
	*Expertise Level*	F_1,113_ = 0.205			*p* = 0.652	ƞp2 = 0.002		*Interaction*		
								F_2,113_ = 2.235	*p* = 0.112	ƞp2 = 0.040
**MVM**	**Expert**	0.070	(0.027)	0.066	(0.015)	0.057	(0.019)	F_2,113_ = 0.773	*p* = 0.464	ƞp2 = 0.014
	**Recreational**	0.056	(0.050)	0.057	(0.025)	0.053	(0.016)			
		F_1,113_ = 2.442			*p* = 0.121	ƞp2 = 0.022		*Interaction*		
								F_2,113_ = 0.220	*p* = 0.803	ƞp2 = 0.004
**FE_D_**	*Expertise Level*	0.490	(0.164)	0.501	(0.102)	0.531	(0.166)	F_2,113_ = 1.241	*p* =0.293	ƞp2 = 0.022
	**Recreational**	0.382	(0.268)	0.498	(0.205)	0.509	(0.149)			
	*Expertise Level*	F_1,113_ = 3.651			*p* = 0.059	ƞp2 = 0.033		*Interaction*		
								F_2,113_ = 0.754	*p* = 0.473	ƞp2 = 0.014
**FE_V_**	**Expert**	1.567	(0.214)	1.591	(0.167)	1.627	(0.166)	F_2,113_ = 10.525	*p* < 0.001	ƞp2 = 0.163
	**Recreational**	1.113	(0.575)*	1.633	(0.186)^A^	1.608	(0.232)^A^			
	*Expertise Level*	F_1,113_ = 5.860			*p* = .017	ƞp2 = .051		*Interaction*		
								F_2,113_ = 7.354	*p* = 0.001	ƞp2 = 0.120
**DFA_D_**	**Expert**	1.035	(0.134)	0.969	(0.188)	1.001	(0.133)	F_2,113_ = 2.798	*p* =0.065	ƞp2 = 0.049
	**Recreational**	1.146	(0.162)*	1.057	(0.203)	1.047	(0.126)			
	*Expertise Level*	F_1,113_ = 7.449			*p* = 0.007	ƞp2 = 0.065		*Interaction*		
								F_2,113_ = 0.484	*p* = 0.618	ƞp2 = 0.009
**DFAv**	**Expert**	0.851	(0.101)	0.851	(0.085)	0.815	(0.087)	F_2,113_ = 2.665	*p* = 0.074	ƞp2 = 0.047
	**Recreational**	0.859	(0.075)	0.899	(0.090)	0.836	(0.107)			
	*Expertise Level*	F_1,113_ = 2.065			*p* = 0.154	ƞp2 = 0.019		*Interaction*		
								F_2,113_ = 0.392	*p* = 0.677	ƞp2 = 0.007

MRE: mean radial error (m); ball speed (km/h); BVE: bivariate variable error (m); MVM: mean velocity magnitude (m/s); COP: center of pressure; FED: fuzzy entropy of the displacement of COP (unitless); FEV: fuzzy entropy of the velocity of COP (unitless); DFAD: detrended fluctuation analysis of the displacement of COP (unitless); DFAV: detrended fluctuation analysis of the velocity of COP (unitless). * Differences with respect to the expert group. ^A^ Significant differences with respect to U12. ^B^ Significant differences with respect to U16.

**Table 3 entropy-22-00822-t003:** Correlations between all variables for all the subjects.

	Ball speed	BVE	MVM	FE_D_	FE_V_	DFA_D_	DFA_V_
MRE (m)	−0.042	0.050	−0.196*	−0.264**	−0.068	0.159	−0.121
Ball speed (km/h)	--	−0.107	0.092	0.237*	0.303**	−0.211*	−0.042
BVE (m)	--	--	0.392**	−0.121	−0.203*	0.163	0.034
MVM (m/s)	--	--	--	0.745**	0.321*	−0.398**	−0.063
FE_D_	--	--	--	--	0.607**	−0.645**	−0.106
FE_V_	--	--	--	--	--	−0.440**	−0.200*
DFA_D_	--	--	--	--	--	--	0.193*
* *p* < .05; ** *p* < .01							

MRE: mean radial error (m); ball speed (km/h); BVE: bivariate variable error (m); MVM: mean velocity magnitude (m/s); FE_D_: fuzzy entropy of the displacement of COP (unitless); FE_V_: fuzzy entropy of the velocity of COP (unitless); DFA_D_: detrended fluctuation analysis of the displacement of COP (unitless); DFA_V_: detrended fluctuation analysis of the velocity of COP (unitless).

**Table 4 entropy-22-00822-t004:** Correlations between all variables according to age ranges.

U12		Ball speed (km/h)	BVE (m)	MVM (m/s)	FE_D_	FE_V_	DFA_D_	DFA_V_
	MRE	−0.112	0.178	−0.176	−0.248	−0.115	0.163	−0.205
	Ball speed	--	0.166	0.157	0.184	0.380*	−0.350*	−0.077
	BVE	--	--	0.300	−0.043	−0.046	0.128	0.065
	MVM	--	--	--	0.851**	0.423**	−0.506**	−0.016
	FE_D_	--	--	--	--	0.678**	−0.694**	−0.073
	FE_V_	--	--	--	--	--	−0.657**	−0.162
	DFA_D_	--	--	--	--	--	--	0.116
U16		Ball speed (km/h)	BVE (m)	MVM (m/s)	FE_D_	FE_V_	DFA_D_	DFA_V_
	MRE	0.189	0.320	0.034	−0.248	−0.133	0.249	−0.007
	Ball speed	--	0.328	0.427*	0.228	−0.061	0.347	−0.050
	BVE	--	--	0.593**	−0.106	−0.352	0.077	0.247
	MVM	--	--	--	0.706**	0.263	−0.534**	−0.389*
	FE_D_	--	--	--	--	0.582**	−0.767**	−0.314
	FE_V_	--	--	--	--	--	−0.228	−0.298
	DFA_D_	--	--	--	--	--	--	0.275
18+		Ball speed (km/h)	BVE (m)	MVM (m/s)	FE_D_	FE_V_	DFA_D_	DFA_V_
	MRE	−0.156	−0.157	−0.357*	−0.343*	−0.085	0.176	−0.055
	Ball speed	--	−0.281	0.015	0.227	0.050	−0.102	−0.308*
	BVE	--	--	0.542**	−0.146	−0.180	0.114	0.041
	MVM	--	--	--	0.613**	0.225	−0.261	−0.068
	FE_D_	--	--	--	--	0.444**	−0.482**	−0.030
	FE_V_	--	--	--	--	--	0.000	−0.294
	DFA_D_	--	--	--	--	--	--	0.220
* *p* < .05; ** *p* < .01								

MRE: mean radial error (m); ball speed (km/h); BVE: bivariate variable error (m); MVM: mean velocity magnitude (m/s); FE_D_: fuzzy entropy of the displacement of COP (unitless); FE_V_: fuzzy entropy of the velocity of COP (unitless); DFA_D_: detrended fluctuation analysis of the displacement of COP (unitless); DFA_V_: detrended fluctuation analysis of the velocity of COP (unitless).

**Table 5 entropy-22-00822-t005:** Correlations between all variables according to skill level.

Expert							
	Ball speed	BVE	MVM	FE_D_	FE_V_	DFA_D_	DFA_V_
MRE (m)	0.216	−0.166	−0.205	−0.220	0.026	0.104	0.222
Ball speed (km/h)	--	−0.350**	−0.214	0.008	−0.097	−0.102	−0.076
BVE (m)	--	--	0.595**	−0.125	−0.232	0.268*	0.164
MVM (m/s)	--	--	--	0.629**	0.190	−0.176	0.259*
FE_D_	--	--	--	--	0.480**	−0.496**	0.176
FE_V_	--	--	--	--	--	−0.093	−0.286*
DFA_D_	--	--	--	--	--	--	−0.074
Recreational							
	Ball speed	BVE	MVM	FE_D_	FE_V_	DFA_D_	DFA_V_
MRE (m)	0.008	0.225	−0.165	−0.252	−0.051	0.140	−0.071
Ball speed (km/h)	--	0.067	0.132	0.230	0.351**	−0.237	−0.328*
BVE (m)	--	--	0.290*	−0.121	−0.217	0.081	−0.089
MVM (m/s)	--	--	--	0.793**	0.332*	−0.514**	−0.255
FE_D_	--	--	--	--	0.640**	−0.724**	−0.282*
FE_V_	--	--	--	--	--	−0.561**	−0.151
DFA_D_	--	--	--	--	--	--	0.410**
* *p* < .05; ** *p* < .01							

MRE: mean radial error (m); ball speed (km/h); BVE: bivariate variable error (m); MVM: mean celocity magnitude (m/s); FE_D_: fuzzy entropy of the displacement of COP (unitless); FE_V_: fuzzy entropy of the velocity of COP (unitless); DFA_D_: detrended fluctuation analysis of the displacement of COP (unitless); DFA_V_: detrended fluctuation analysis of the velocity of COP (unitless).

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
