# Peer review of "Functional Variability in Team-Handball Players during Balance Is Revealed by Non-Linear Measures and Is Related to Age and Expertise Level"

_entropy, 2020, doi:10.3390/e22080822_

Round 1

Reviewer 1 Report

Journal: Entropy (ISSN 1099-4300)

Manuscript ID: entropy-860286

Title: Functional Variability in Team-Handball Players during Balance Is Revealed by Non-Linear Analysis And It Is Related To Age And Expertise Level

General comments

Thank you for giving me the opportunity to review this paper.

This study aim to assess the relationship between balance ability and sport performance of the throwing in team-handball players of different ages and levels of expertise. In fact, the non-linear approach revealed that functional variability is related to age and expertise level and have little doubt that this study would be of interest to readers of this journal. The methods of assessment and analysis are appropriate, and the results have been interpreted correctly. The conclusion is a fair reflection and interpretation of the results, but what is lacking is the practical implications.

In accordance, some minor issues need to be considered to implement the quality of their work.

The comments are included below.

Specific comments

Abstract

1/ Clarify. Male, female or male and female team-handball players? (also in method)

2/ Suggestion: “... experienced team-handball players exhibited better balance performance, and it was related to maturation of the motor system more than to sport performance level.” // “.., and it can be related…” or “…, and it seem to be related…”

Introduction

1/ Paragraph 2, Line 4: “… studies that have tried…” // References?

Method / Subjects

1/ “None of them had previous experience in the balance task used in this study”// this is a possible limitation in the study and need to be addressed in the discussion (i.e., in my opinion, previous test was recommended in order to familiarization). // Present this limitation of the study in discussion.

Discussion

1/ Just a note: Non-linear measures seems to be helpful to quantify how motor behaviour changes over time. Nevertheless, the main objective of the study is to assess the relationship between balance ability and sport performance of the throwing in team-handball players of different ages and levels of expertise. In other words, the non-linear approach to balance assessment is “only” a methodological option of the authors to clarify the relationship between balance ability and sports performance. Sometimes it seems that this is the main goal of the paper.

2/ Provide some practical implications.

References

Check all the references (after revisions).

Tables

Table 1.: Elite? Use the same expression (i.e.: EXPERT).

Author Response

The authors would like to thank you for your advice and recommendations, which in our view have contributed to improve the paper.

Please, see below the point-by-point responses to your comments:

Specific comments:

Abstract:

1/ Clarify. Male, female or male and female team-handball players? (also, in method)

Authors: The participants’ gender has been added in the abstract and method.

2/ Suggestion: “... experienced team-handball players exhibited better balance performance, and it was related to maturation of the motor system more than to sport performance level.” // “..., and it can be related…” or “…, and it seems to be related…”

Authors: Thank you very much for the suggestion. The reviewed manuscript has been modified as follows: “In conclusion, the better performance exhibited (i.e., lower sway) by experienced team-handball players seems to be related to maturation of the motor system more than to sport performance level.”

1/ Paragraph 2, Line 4: “… studies that have tried…” // References?

Authors: References regarding this sentence have been added at the end of the sentence: “Studies have indicated that sport training improves sensorimotor performance and postural control [1, 2]”.

Method / Subjects

1/ “None of them had previous experience in the balance task used in this study”// this is a possible limitation in the study and need to be addressed in the discussion (i.e., in my opinion, previous test was recommended in order to familiarization). // Present this limitation of the study in discussion. 

Authors: The authors are aware that the number of repetitions and the lack of experience in the task could affect to reliability of the measurements. Nevertheless, it was a requirement to be include in the study to avoid that participants could have high balance level because of his\her previous experience and not because of handball training. This issue has been included as a limitation in the manuscript

Discussion

1/ Just a note: Non-linear measures seems to be helpful to quantify how motor behavior changes over time. Nevertheless, the main objective of the study is to assess the relationship between balance ability and sport performance of the throwing in team-handball players of different ages and levels of expertise. In other words, the non-linear approach to balance assessment is “only” a methodological option of the authors to clarify the relationship between balance ability and sports performance. Sometimes it seems that this is the main goal of the paper.

 Authors: We want to thank you for the thought. We agree that the non-linear approach to balance assessment provide another methodological option. In the introduction section, it has been discussed that non-linear tools it could be a better approach to analyze the motor control processes in balance tasks related to sport performance and experience.

2/ Provide some practical implications.

Authors: A new section named Practical Implications has been added after the Discussion.

References:

1/ Check all the references (after revisions).

Authors: All the references have been checked after addressing all the reviewers´ comments.

Tables:

Table 1.: Elite? Use the same expression (i.e.: EXPERT).

Authors: The change has been made.

Reviewer 2 Report

My comments can be found in attachment.

Author Response

The authors would like to thank you for your advice and recommendations, which in our view have contributed to improve the paper.

The authors’ responses to each comment are presented below, and changes in the manuscript are presented in red. In addition, the reviewed manuscript has been submitted to Altair K. Fanto Proof-Reading-Service (NIF: X0692378D; C/ La Goleta, 6; C.P.: 03540, Alicante, Spain) for editing and proofreading.

SPECIFIC COMMENTS:

Title: I suggest to drop “It” from the title.

Authors: The title has been modified according to this suggestion.

Abstract:

  1. “Postural control in balance tasks has been investigated in relation to sport skill level, reporting controversial results. The skill level and the age of the performers have been suggested as mediators of this relationship, but they have not been addressed yet.” Do you mean ‘conflicting results’? How can ‘skill level’ be a mediator in the relationship between skill level and postural control? Please rephrase this.

Authors: We understand that the term controversial could not be clear. With this expression, we meant that the relationship between balance and sport performance in different athletic skills has not always been observed. They are not necessarily conflicting, but the conclusions are not clear. We have rephrased the sentence in abstract and introduction to clarify.

  1. “Players who throwed threw with less accuracy …”

Authors: The change has been made.

  1. “In conclusion, experienced team-handball players exhibited better balance performance, …” Was balance ‘performance’ assessed? The non-linear analysis reveals postural control principles, but performance refers more to measures like time until failure or number of stumbles, etc.

Authors: We agree with the assumption that no-linear measures reveal postural control principles. Nevertheless, this affirmation refers to the results observed in the Bivariate Variable Error that quantify the oscillation/sway magnitude of the COP displacement (i.e., average value of the absolute distance to each participant’s own midpoint). As it is explained in the Method section, this measure, as well as, the Mean Velocity Magnitude were used to assess balance performance. We have modified the sentence to clarify its meaning.

Introduction:

  1. “… the findings seem to be controversial [4].” Do you mean conflicting? From the following sentences, it seems that it should be conflicting.
  2. “Another possible reason for the previous controversial results may be the variable used to assess balance performance.” Same as before.
  3. “Due to the controversial results reported by the literature, …” Same as before.

Authors: We want to thank this suggestion. Those sentences have been rephrased.

  1. “…, the aim of this study was to assess the relationship between balance ability and sport performance of the throwing (defined as throwing speed and accuracy) in team-handball

players of different ages and levels of expertise …” Suggest to rephrase as indicated.

Authors: The sentence has been rephrased following the suggestion.

  1. “… to analyze how relevant balance control seems to be.” This is strangely formulated. If you would find a weak relationship, does that imply that balance control is irrelevant? Likely not, so please rephrase this.

 Authors: Sentence have been rephrased and changed to “expertise to analyze the relevance of balance control in throwing performance”.

Methods

  1. “… a rigid unstable surface …” Please give more details on what surface was used. The reader must have an idea of how unstable this is (maybe add pictures or movies as online material?).

Authors: We have added the following details in the reviewed manuscript:

“Participants were asked to stand “as still as possible” [24] on an unstable surface for one minute. The unstable surface consisted in a rigid wooden platform (diameter: 55 cm) affixed to the flat surface of a polyester resin hemisphere (diameter of hemisphere: 35 cm; height of the platform relative to the bottom of the hemisphere: 12 cm).

  1. “… during one minute.” Write out in full please.

 Authors: This change has been made.

  1. “FE typically returns values that indicate the degree of irregularity in the signal.” Please give a bit more detail on how FE can be interpreted in the context of COP signals. Why was this type of entropy chosen (versus other choices of sample entropy, approximate entropy, permutation entropy, spectral entropy, …)? It’s not necessary to discuss these alternatives, but information of why fuzzy entropy is chosen is important, I think.

Authors: All the information the reviewer required and some information about how FE was computed have been added:

“FE typically returns values that indicate the degree of irregularity in the signal. It was calculated according to Chen, Wang [29]. This measure computes the repeatability of vectors of length m and m + 1 that repeat within a tolerance range of r of the standard deviation of the time series. Thus, low FE values represent a high repeatability within the time series as similar vectors of length m remain similar for the next + 1 points. Conversely, high FE values represent a low repeatability within the time series as similar vectors of length m do not remain similar for the next + 1 points. To calculate this measure, the following parameter values were used: vector length, m = 2; tolerance window, r = 0.2*SD; and gradient, n=2 [29]. Although there are many entropy measures, FE was used because it displays a stronger relative consistency, less dependency on data length, free parameter selection and more robustness to noise than other measures like Sample Entropy [30]”.

  1. “Multivariate analysis was used to analyze the differences between ages and skill level.” Please be more specific (e.g. two-way ANOVAs with main effects of age and skill level and age*skill interaction effect). Also, only univariate analyses are reported in the results section; what multivariate analyses were conducted (MANOVA)?

Authors: We like to thank the reviewer to point out this mistake. In fact, univariate analyses were only carried out in this study. The information about statistical procedures has been enhanced and clarified as follows:

“Two-way independent measures ANOVAs were carried out to assess between group differences in all parameters, being age (U12, U16 and +18) and skill level (expert and recreational) the between-subjects’ factors.”

Results:

  1. Table 2 caption: drop ‘differences’ or rephrase the caption (no differences are shown in the table).

Authors: The word “differences” has been removed.

  1. “The results showed clearly how that experts threw faster than recreational players, finding differences between expertise levels for all age ranges (p<.001).”

Authors: The change has been made.

  1. “Players who served threw faster (higher ball speed) …”

Authors: The change has been made.

  1. “However, players who served threw faster (higher ball speed) …”

Authors: The change has been made.

Discussion:

  1. “Previous studies have showedn controversial results in the relationship …” Controversial should be conflicting?

Authors: The change has been made, and “controversial” has been replaced by “unclear” in the manuscript.

  1. “With regard to correlational analysis, the team-handball performance variables (accuracy and ball speed) were not related to each other.” I guess this is also related to the data reduction that was done. A correlation at population level was examined between mean throwing speed and mean error (averaged within subjects). When a correlation would be examined within subjects between throwing speed and error, a negative relationship is more likely. Can the authors verify this please?

Authors: The correlations for each specific group (Expert +18; Expert U16, Expert U12, Recreational +18; Recreational U16 and Recreational U12) and whiting-subjects were computed, but the correlation between speed and accuracy did not appear in any of them.

  1. “On the other hand, players who served threw faster displayed COP movements …”

Authors: The change has been made.

Reviewer 3 Report

Summary:

In their study, the authors analyzed the relationship between balance ability and performance in throwing balls (in a handball sport setting), taking into account the age and the level of expertise of their participants. They further reported that although the experts threw faster but not their accuracy was not higher than the recreational players. Moreover, they observed that the age group 18+ showed higher balance than those in age group under 12. On the other hand, this difference was not related to their skill level.

Major concern:

1) It is not clear what the authors meant by “ Non-Linear Analysis ”when referring to the analyses presented. One part of the analyses were based on correlation analysis. The other part referred to the term “multivariate analysis” but the manuscript failed to explain clearly what the authors meant by multivariate analysis. Does it refer to using multi-feature measurements that the authors collected? What type of multivariate analysis? What similarity/distance measure was adapted?

Reading through the Discussion Section, it appears that the authors use the term non-linear with regards to the calculated features. If this is the case, it should be clearly mentioned in the manuscript since the analyses do not adapt any non-linear steps and therefore the reference to this term and in particular in the title of the manuscript is quite misleading.

2) Effect sizes are not reported and it is also not clear how the F-statistics are computed.

3) The methods are not sufficiently explained and therefore it is quite difficult to comprehend what the calculated measures in this study actually represent/quantify.

4) The writing of the manuscript is not easy to follow and would certainly benefit from a thorough auditing and a proofread.

5) In the Results Section, the authors mentioned “On the other hand, players who threw faster (higher ball speed) displayed COP movements that were more irregular (higher FED and FEV) and less autocorrelated (lower DFAD).” however, it appears that no autocorrelation analyses were reported.

6) In the Results Section, the authors mentioned “When the sample was split by age range, ...” why factor (e.g., age, expertise, gender) analysis was not used? The manuscript reports that “Normality was evaluated using the Kolmogorov-Smirnov test with the Lilliefors correction.” which I assume to imply that the data was normally distributed (however, no result is provided). Also, the gender ratio of the participants is not provided.

7) In Discussion Section, the authors first noticed that “However, MRE values did not differ between age ranges or skill level groups. ” but later mentioned that “The variables that described the variability structure of the COP during the balance task were more sensitive to the differences not only between age ranges but between skill levels. ” This appears inconsistent/contradictory as one part of the results is interpreted as not related to the skill while the other attribute the difference to expertise. Please provide further clarification.

8) The authors referred to “autocorrelation” throughout the manuscript, suggesting its significant for interpreting the observations. However, no autocorrelation analyses were reported. This requires further clarification and the manuscript needs to be modified/extended accordingly.

9) In the Discussion Section, the authors mentioned that “Yet, we did not find any relationship between performance variables when we analyzed only the recreational group. ” I strongly suggest that the authors consider applying factor analysis on their data (e.g., factors: age, expertise) other than analyzing them within each group separately. I suspect that one reason behind non-significant differences between the experts and novices could be due to discarding these factors during the analyses.

Further Comments:

Please use consistent tense form and avoid switching between present and past tenses.

Abstract, Results Section:

Please define and/or provide a short description for the term “U12” so the readers not familiar with this topic.

throwed ” to “threw”

balance performance, and it was related to maturation of the motor system more than to sport performance level.” to “balance performance which was more related to maturation of the motor system than to sport performance level.”

Introduction

relationship of balance with athletic” to “relationship between the balance and athletic”

seem to be controversialto “appear to be controversial”

hockey skatingto “ice hockey ”

they indicated that this relation appears justto “they identified that this relation was only present”

control better identified ” to “control that are better identified ”

suggested that, in sports like this one, ” to “suggested that in these type of sports”

have not been assessed yet. ” to “have not been assessed to this date.”

to analyze how relevant balance control seems to be. ”: This is not clear. Please clearly state what was the hypothesis of this study.

Subjects: (It would be more informative to change this heading to “Participants”)

prior testing” to “prior to the experiment”

Please provide descriptions for “U12” and “U16.” I noted that the author(s0 provided this information after Table 1. However, it helps the readers better follow the content if it is provided earlier and before referring to these terms.

who were considerto “who were considered”, also, please move the “*” and “#” points to the table’s caption.

who have the followed requirements ” to “who satisfied the following requirements”

It is in “*” that the authors first referred to “Spanish Handball Federation.”Please clarify this information at start of the “Subject” (e.g., when stating the demography of the participants). Please apply these corrections/modifications to both, “*” and “#”

Experimental Procedure and Data Collection

during one min.to “for one minute.”

during in each throwto “during each throw”

Caption of Figure 1 is missing.

Data analysis and reduction

displacement trajectories: Bivariate Variable Error (BVE) and Mean Velocity Magnitude (MVM). This part is not clear. Are these the quantities measured and used for further analyses? Please provide further clarification/explanation.

were calculated over the signal and calculated according to Prieto, Myklebust ” to “were calculated over the signal using the approach by Prieto, Myklebust ”

velocity were Fuzzy Entropy (FED and FEV) and Detrended Fluctuation Analysis (DFAD and DFAV). ” It is not clear which velocities the authors referred to. Please also explain what are FE_D and FE_V. Same for FDA.

according to the procedure of ..to “according to the procedure by...”

Also, in the heading, it is not clear what the term “reduction” refers to.

Results

was the only one ” to “was the only value” OR “was the only parameter”

In Tables, the effect sizes are missing. It is important to report the effect sizes to allow for more informed interpretation of the observed differences.

Author Response

The authors would like to thank you for your advice and recommendations, which in our view have contributed to improve the paper.  

The authors’ responses to each comment are presented below, and changes in the manuscript are presented in red. In addition, the reviewed manuscript has been submitted to Altair K. Fanto Proof-Reading-Service (NIF: X0692378D; C/ La Goleta, 6; C.P.: 03540, Alicante, Spain) for editing and proofreading.

Summary: 

In their study, the authors analyzed the relationship between balance ability and performance in throwing balls (in a handball sport setting), taking into account the age and the level of expertise of their participants. They further reported that although the experts threw faster but not their accuracy was not higher than the recreational players. Moreover, they observed that the age group 18+ showed higher balance than those in age group under 12. On the other hand, this difference was not related to their skill level.

Major concern:

  1. It is not clear what the authors meant by “Non-Linear Analysis” when referring to the analyses presented. One part of the analyses was based on correlation analysis. The other part referred to the term “multivariate analysis” but the manuscript failed to explain clearly what the authors meant by multivariate analysis. Does it refer to using multi-feature measurements that the authors collected? What type of multivariate analysis? What similarity/distance measure was adapted?

Reading through the Discussion Section, it appears that the authors use the term non-linear with regards to the calculated features. If this is the case, it should be clearly mentioned in the manuscript since the analyses do not adapt any non-linear steps and therefore the reference to this term and in particular in the title of the manuscript is quite misleading.

Authors: The authors completely agree with the reviewer´s comment. When we talk about non-linear analysis, we refer to the non-linear tools are used to assess the dynamic of the signal. We did not perform any other non-linear analysis, so we have changed the term of non-linear analysis by non-linear measures along the manuscript.

  1. Effect sizes are not reported and it is also not clear how the F-statistics are computed.

Authors: The Effect sizes have been added in the Tables 2 and 3.

  1. The methods are not sufficiently explained and therefore it is quite difficult to comprehend what the calculated measures in this study actually represent/quantify.

Authors: More detailed information has been added regarding the variables and their computation. All the changes are color-coded in red in the manuscript.

  1. The writing of the manuscript is not easy to follow and would certainly benefit from a thorough auditing and a proofread.

Authors: The reviewed manuscript has been submitted to Altair K. Fanto Proof-Reading-Service (NIF: X0692378D; C/ La Goleta, 6; C.P.: 03540, Alicante, Spain) for editing and proofreading.

  1. In the Results Section, the authors mentioned “On the other hand, players who threw faster (higher ball speed) displayed COP movements that were more irregular (higher FED and FEV) and less autocorrelated (lower DFAD).” however, it appears that no autocorrelation analyses were reported.

Authors: DFA is a non-linear tool that measures the presence of long-term auto-correlations within a time series. Thus, when we talk about autocorrelation we are referring to this tool. We have tried to clarify the meaning of DFA adding detailed information in the method section:

“DFA represents a modification of classic root mean square analysis with random walk to evaluate the presence of long-term correlations within a time series using a parameter referred to as the scaling index, α [31]. The scaling index α corresponds to a statistical dependence between fluctuations at one time scale and those over multiple time scales [32]. This variable was computed according to the procedures by Peng, Havlin [31], using the window range 4 ≤ n ≤ N/10 to maximize the long-range correlations and reduce errors incurred in by estimating α [33]. Different values of α indicate the following: α > 0.5 implies persistence in position (the trajectory tends to remain in its current direction); α < 0.5 implies anti-persistence in position (the trajectory tends to return to where it came from) [34]”.

  1. In the Results Section, the authors mentioned “When the sample was split by age range, ...” why factor (e.g., age, expertise, gender) analysis was not used? The manuscript reports that “Normality was evaluated using the Kolmogorov-Smirnov test with the Lilliefors correction.” which I assume to imply that the data was normally distributed (however, no result is provided). Also, the gender ratio of the participants is not provided.

Authors: The gender of the participants has been added in the abstract and method.

  1. In Discussion Section, the authors first noticed that “However, MRE values did not differ between age ranges or skill level groups. ” but later mentioned that “The variables that described the variability structure of the COP during the balance task were more sensitive to the differences not only between age ranges but between skill levels. ” This appears inconsistent/contradictory as one part of the results is interpreted as not related to the skill while the other attribute the difference to expertise. Please provide further clarification.

Authors: The apparent inconsistency can be due to that we are referring to different variables (throwing accuracy and balance variability structure). MRE is the variable used to assess the accuracy during the handball throwing while the variability structure of the COP is described by the variables FE and DFA. We have mentioned the variables (in brackets) in the latter sentence to clarify.

  1. The authors referred to “autocorrelation” throughout the manuscript, suggesting its significant for interpreting the observations. However, no autocorrelation analyses were reported. This requires further clarification and the manuscript needs to be modified/extended accordingly.

Authors: Please, see above the answer to the comment 5.

  1. In the Discussion Section, the authors mentioned that “Yet, we did not find any relationship between performance variables when we analyzed only the recreational group” I strongly suggest that the authors consider applying factor analysis on their data (e.g., factors: age, expertise) other than analyzing them within each group separately. I suspect that one reason behind non-significant differences between the experts and novices could be due to discarding these factors during the analyses.

Authors: In table 2, you can find the data from the factor analysis performed, indicating main effects of each factor (age, expertise) and interaction effects. We have also added effect size values.

Further Comments:

  1. Please use consistent tense form and avoid switching between present and past tenses.

Authors: The manuscript has been checked to unify the tense form.

Abstract, Results Section:

  1. Please define and/or provide a short description for the term “U12” so the readers not familiar with this topic.

Authors: U12 is used for participants under 12 years old. These terms are explained in the legend of the tables and we have also added the explanation in the abstract and in method section.

  1. “throwed ” to “threw”

Authors: This change has been made along all the manuscript.

  1. “balance performance, and it was related to maturation of the motor system more than to sport performance level.” to “balance performance which was more related to maturation of the motor system than to sport performance level.”

Authors: That sentence has been rephrased.

Introduction:

  1. “relationship of balance with athletic” to “relationship between the balance and athletic”

Authors: The change has been made.

  1. “seem to be controversial” to “appear to be controversial”

Authors: The change has been made.

  1. “hockey skating” to “ice hockey”

Authors: We did not change this term because it is the one used by the authors of the paper.

  1. “they indicated that this relation appears just” to “they identified that this relation was only present”

Authors: The change has been made.

  1. “control better identified” to “control that are better identified”

Authors: The change has been made.

  1. “suggested that, in sports like this one,” to “suggested that in these type of sports”

Authors: The change has been made.

  1. “have not been assessed yet.” to “have not been assessed to this date.”

Authors: The change has been made.

  1. “to analyze how relevant balance control seems to be.”: This is not clear. Please clearly state what was the hypothesis of this study.

Authors: This sentence has been changed in order to clarify it.

Subjects: (It would be more informative to change this heading to “Participants”)

Authors: The change has been made.

  1. “prior testing” to “prior to the experiment”

Authors: The change has been made.

  1. Please provide descriptions for “U12” and “U16.” I noted that the authors provided this information after Table 1. However, it helps the readers better follow the content if it is provided earlier and before referring to these terms.

Authors: We have also added the explanation in the abstract and the first time they are used in the text.

  1. “who were consider” to “who were considered”, also, please move the “*” and “#” points to the table’s caption.

Authors: We have changed “consider” by “considered”. The information regarding “*” and “#” have been moved to the text of the manuscript (participants section) to better clarification for the reader.

  1. “who have the followed requirements” to “who satisfied the following requirements”

Authors: The change has been made.

  1. It is in “*” that the authors first referred to “Spanish Handball Federation” Please clarify this information at start of the “Subject” (e.g., when stating the demography of the participants). Please apply these corrections/modifications to both, “*” and “#”

Authors: We have specified that information about the participants in the text of the method section to avoid confusion with “*” and “#” symbols. The symbols have been removed from the table.

Experimental Procedure and Data Collection:

  1. “during one min.” to “for one minute.”

Authors: The change has been made.

  1. “during in each throw” to “during each throw”

Authors: The change has been made.

  1. Caption of Figure 1 is missing.

Authors: The figure caption has been added.

Data analysis and reduction:

  1. “displacement trajectories: Bivariate Variable Error (BVE) and Mean Velocity Magnitude (MVM).” This part is not clear. Are these the quantities measured and used for further analyses? Please provide further clarification/explanation.

Authors: More detailed information has been added regarding the variables used and how they were computed.

  1. “were calculated over the signal and calculated according to Prieto, Myklebust ” to “were calculated over the signal using the approach by Prieto, Myklebust ”

Authors: The change has been made.

  1. “velocity were Fuzzy Entropy (FED and FEV) and Detrended Fluctuation Analysis (DFAD and DFAV).” It is not clear which velocities the authors referred to. Please also explain what are FE_D and FE_V. Same for FDA.

Authors: This paragraph has been changed to clarify.

  1. “according to the procedure of ..” to “according to the procedure by...”

Authors: The change has been made.

  1. Also, in the heading, it is not clear what the term “reduction” refers to.

Authors: The word reduction has been removed.

Results:

  1. “was the only one” to “was the only value” OR “was the only parameter”

Authors: The word “one” has been changed by variable.

  1. In Tables, the effect sizes are missing. It is important to report the effect sizes to allow for more informed interpretation of the observed differences.

Authors: The effect sizes have been added to the Tables 2 y 3.

Round 2

Reviewer 2 Report

The authors have answered adequately to all my queries and I am happy to recommend accepting the paper. Congratulations on a fine article!

Author Response

The authors would like to thank you for your advice and recommendations, which in our view have contributed to improve the paper.

Reviewer 3 Report

Thank you very much for addressing my comments. The manuscript has been substantially improved. I think it can be published after a minor modification. 

In Table 2, the caption reads "Main effects and interaction effects (F, p values and Effect Size) are also presented." However, it seems the effect sizes have not been added to this Table's entries. Please include them.

Author Response

Reviewer's comment:

In Table 2, the caption reads "Main effects and interaction effects (F, p values and Effect Size) are also presented." However, it seems the effect sizes have not been added to this Table's entries. Please include them.

Authors: The Effect sizes have been added.

The authors would like to thank you for your advice and recommendations, which in our view have contributed to improve the paper.